# The Scientific Basis for Occupational Exposure Limits for Hydrogen Sulphide—A Critical Commentary

**DOI:** 10.3390/ijerph18062866

**Published:** 2021-03-11

**Authors:** Mark Elwood

**Affiliations:** Department of Epidemiology & Biostatistics, School of Population Health, University of Auckland, Private Bag 92019, Auckland Mail Centre, Auckland 1142, New Zealand; mark.elwood@auckland.ac.nz

**Keywords:** hydrogen sulphide, occupation, health, exposure limits, review

## Abstract

Objectives: Occupational exposure limits for hydrogen sulphide (H_2_S) vary considerably; three expert group reports, published from 2006 to 2010, each recommend different limits. Some jurisdictions are considering substantial reductions. Methods: This review assesses the scientific evidence used in these recommendations and presents a new systematic review of human studies from 2006–20, identifying 33 studies. Results: The three major reports all give most weight to two sets of studies: of physiological effects in human volunteers, and of effects in the nasal passages of rats and mice. The human studies were done in one laboratory over 20 years ago and give inconsistent results. The breathing style and nasal anatomy of rats and mice would make them more sensitive than humans to inhaled agents. Each expert group applied different uncertainly factors. From these reports and the further literature review, no clear evidence of detrimental health effects from chronic occupational exposures specific to H_2_S was found. Detailed studies of individuals in communities with natural sources in New Zealand have shown no detrimental effects. Studies in Iceland and Italy show some associations; these and various other small studies need verification. Conclusions: The scientific justification for lowering occupational exposure limits is very limited. There is no clear evidence, based on currently available studies, that lower limits will protect the health of workers further than will the current exposure limits used in most countries. Further review and assessment of relevant evidence is justified before exposure limits are set.

## 1. Introduction

Hydrogen sulphide (molecular formula H_2_S; CAS no. 7783-06-4; 1 ppm = 1.394 mg/m^3^ and 1 mg/m^3^ = 0.717 ppm at 25 °C) is a colourless gas with a characteristic odour of ‘rotten eggs’. It is present in volcanic gases and other geothermal emissions, and produced by decomposition in sewers and stagnant water. Occupational exposure occurs in the ‘sour gas’ segment of the natural gas industry, processing gas with high sulphur concentrations. Hydrogen sulphide can be generated in petroleum refineries, natural gas plants, petrochemical plants, coke oven plants, kraft paper mills, viscose rayon manufacture, sulphur production, iron smelters, food processing plants, tanneries, landfills, manure treatment plants and waste water treatment plants [1,2,3,4,5].

Hydrogen sulphide has a simple physiological effect, that of its unpleasant smell. Some communities live with this, the largest being Rotorua, a city of about 60,000 population situated on a geothermal field in New Zealand. A report shows the threshold limit of detection as 0.7 μg/m^3^ (0.5 ppb), although Rotorua residents can tolerate 100 times this threshold routinely [6]. Although unpleasant, irritating, and regarded as unacceptable in the general population, this effect is accepted in occupational settings, as even the lowest suggested exposure limits would not protect against it. Sensitivity to odours in general decreases rapidly with repeated exposures (olfactory fatigue) and is affected by psychological state and bias, which in turn can increase subjective reported symptoms; but odour detection does not imply any detrimental health effect [7].

At the other extreme, hydrogen sulphide can cause death and severe illness in high concentrations [4], well above even the highest of currently accepted exposure limits. High exposures can cause ‘olfactory paralysis’, a neurotoxic effect giving loss of smell [8]. H_2_S has been used in suicide [9].

Occupational exposure limits for H_2_S have been produced by three major groups: the ACGIH (American Conference of Government Industrial Hygienists), SCOEL (Scientific Committee on Occupational Exposure Limits) in Europe, DECOS (Dutch Expert Committee on Occupational Standards) in The Netherlands. These limits differ. Some countries, such as New Zealand, have recently recommended reducing their current limits to those recommended by ACGIH [1].

The present report considers the recommendations of major groups concerning occupational exposure to H_2_S and the scientific evidence used to develop these, and addresses the key question of whether workers’ health will be better protected by the adoption of lower limits. 

## 2. Materials and Methods

The three expert reports and the studies cited in them were assessed. A further search was made in PubMed and Embase for publications using the terms ‘hydrogen sulphide/sulfide’, and ‘occupation’ or ‘community’ or ‘population’ or ‘ambient’, and ‘human’, from 1 Jan 2006 to 31 Dec 2018, published in English or with an abstract in English. Of 404 citations screened, 254 dealt with other exposures such as carbon disulphide or issues other than health, 36 only with high dose acute exposures, 11 with physiological mechanisms, 49 only with exposures, not outcomes, and 23 were review articles. This left 31 new original studies of H_2_S exposures and health outcomes, 7 of occupational and 24 of community exposures. These were considered in detail and are reviewed here. This literature search was updated in February 2021 and 2 further studies added. 

## 3. Results

### 3.1. Occupational Exposure Limits

Occupational exposure limits for hydrogen sulphide vary considerably (Table 1). Of 30 jurisdictions given in [10], all but 2 have eight hour limits (8-h working day average airborne concentration) of from 5 to 10 ppm (7 to 14 mg/m^3^), and short-term limits (usually 15-min time weighted average exposure) of 10 to 20 ppm (14 to 30 mg/m^3^). Two jurisdictions have lower 8-hr limits: The Netherlands, 1.7 ppm, and Spain, 1 ppm. Spain has a 5 ppm short-term limit, as has Austria, as their short and long-term limits are the same. 

From 2006 to 2010, three expert groups reviewed the relevant scientific evidence and produced differing recommendations (Table 2). The ACGIH (American Conference of Government Industrial Hygienists) in 2010 [2] proposed levels of 1 ppm long-term and 5 ppm short-term. SCOEL (Scientific Committee on Occupational Exposure Limits) in Europe in 2007 [3] recommended levels of 5 ppm long-term and 10 ppm short-term, and DECOS (Dutch Expert Committee on Occupational Standards) in The Netherlands in 2006 [5] and in 2010 [11] recommend 1.6 ppm long-term, with no recommendation for short-term limits. 

Therefore, the evidence relevant to occupational guidelines is evidence of effects above the minimum recommended levels of 1 ppm long-term and 5 ppm short-term, but below the levels of 10 ppm and 20 ppm, respectively, used in some jurisdictions at present.

The clearest evidence to justify lowering exposure limits would be good evidence of detrimental health effects in workers who have been exposed to H_2_S at levels within the existing guidelines. Such evidence would be directly relevant to the potential benefits of reducing exposure limits. Indeed, only such evidence would reliably show if there would be health benefits from reducing limits. 

### 3.2. Scientific Evidence Used in Setting Recommended Limits

#### 3.2.1. Human Studies: Studies of Occupational Exposures and Health Outcomes

Studies, especially long-term studies, of workers exposed to H_2_S in their occupations would be particularly valuable. However, in such studies exposures to H_2_S are usually accompanied by other potentially noxious exposures, and also measurement of H_2_S concentrations is usually lacking.

Studies of eye irritation and occupational exposures are noted in the SCOEL report [3]. In the viscose rayon industry, eye irritation (spinners’ eye) was reported in 1939 [12], but linked to H_2_S and to CS_2_, and only at over 20 ppm. In a later study of 123 workers [12], no eye effects were seen at 3.6 ppm, but some observed at 6.4 ppm. The SCOEL report concludes that all these workers were also exposed to CS_2_ gas, and that no effects of H_2_S alone at levels below 20 ppm, 28 mg/m^3^ have been documented (page 7). 

The SCOEL report also notes studies reporting lung function impairment, but with no estimates of exposure levels. Jappinen [13] studied 26 pulp mill workers exposed to under 10 ppm H_2_S, finding no respiratory effects.

Jappinen [14] assessed cardiovascular mortality in three pulp and paper mills in Finland using sulphite processes. National death rates were used for comparison. Amongst men exposed to H_2_S and organic sulphides, there was a significant excess of cardiovascular deaths (37 deaths, standardised mortality ratio (SMR) 150, 95% confidence interval (CI) 105–206), due to an excess of coronary deaths which was not quite significant (25 deaths, SMR 150, CI 98–222). However, the increased risk was lower in those with longer exposure (5+ years compared to <5 years). Workers in the study had higher smoking rates than the general population, and there was no information on other risk factors. There was no increase in total deaths (55 deaths, SMR 107, CI 81–139). Exposures to H_2_S and to organic sulphides could not be separated. A study of these mills many years later showed concentrations of up to 20 ppm H_2_S, up to 15 ppm methyl mercaptan, and dimethyl disulphide up to 1.5 ppm [15]. 

Carcinogenic effects of H_2_S along with other exposures have been addressed in studies of pulp and paper and viscose rayon manufacturing workers, and have been reviewed and assessed as giving ‘inadequate evidence of carcinogenicity’ by the International Agency for Research on Cancer [16]. This report does not consider H_2_S alone.

In a large study of female workers in petrochemical plants in China, Xu [17] found an increased rate of spontaneous abortions in those reporting exposures to H_2_S. No data on exposure levels is given. However, many associations were assessed and increased risks were found for most chemical exposures, being significant for benzene and gasoline. 

#### 3.2.2. Experimental Studies of Human Volunteers under Laboratory Conditions

Studies have been performed with controlled inhalation of H_2_S concentrations of 0.5, 2, 5, or 10 ppm for 15 to 30 min during exercise. As these experiments deal with precisely measured exposure levels, they are given prominence in these assessments. 

These studies were published by one laboratory group in Canada between 1991 and 1997. The studies have not been replicated or confirmed by any other laboratory. 

In the first of these papers [18], 16 young male volunteers were exposed to 0 (control), 0.5, 2.0, and 5.0 ppm H_2_S during exercise up to maximal levels. There were no effects on heart rate or expired ventilation. At the highest exposure, 5 ppm, and the maximal exercise level, maximum oxygen uptake (VO_2_) was significantly increased, and blood lactate concentrations increased significantly. Despite this, the maximal power output of the subjects was not significantly changed. The authors state: “healthy young male subjects could safely exercise at their maximum metabolic rates while breathing 5.0 ppm H_2_S without experiencing a significant reduction in their maximum physical work capacity during short-term incremental exercise.”

In a second, larger, study using the same methods [19], 13 men and 12 women were exposed to 5 ppm H_2_S while exercising to submaximal levels. The previous results were not confirmed. There were no significant effects on any metabolic, cardiovascular, or arterial blood parameter, or on perceptual responses (rating of perceived exertion) in either sex. None of the subjects reported any adverse health effects. There was no change in oxygen uptake or blood lactate levels, unlike in the previous study at maximal exercise levels.

In a further paper from the same study [20], with 13 men and 12 women at 5 ppm H_2_S, muscle biopsies were analysed for markers of anaerobic and aerobic metabolism: lactate, lactate dehydrogenase, citrate synthase, and cytochrome oxidase. Citrate synthase concentration decreased significantly after H_2_S exposure in male subjects, but not in female subjects. There were no other significant results. As noted below, this change in citrate synthase was not confirmed in a later study at 10 ppm exposure [21].

Further papers deal with exposures of 10 ppm. In a 1996 study [22], there were 19 volunteer subjects; 9 men, average age 24.7, and 10 women, average age 21.8. A large range of measures of pulmonary function were assessed. There were no effects seen. None of the subjects experienced any signs and symptoms as a result of H_2_S exposure. 

A further study, published in 1997, assessed cardiovascular, metabolic and biochemical responses to 10 ppm H_2_S [21] on 28 volunteers, 15 men and 13 women. Again, a large number of physiological parameters were assessed. There were no effects seen in five factors related to arterial blood. There were no significant changes in measurements in muscles, assessed by a muscle biopsy (including finding no change in citrate synthase, in which changes had previously been reported after 5 ppm exposure [20]). There were no effects on cardiovascular responses such as heart rate and blood pressure.

However, VO_2_, a measure of oxygen uptake, showed a significant decrease. The authors mention that this differs from their earlier study where 5 ppm exposure at the same exercise level as in this study gave no change in VO_2_ [19]. They do not comment on their other previous study at 5 ppm and maximal exercise, which had shown an increase in VO_2_ [18]. VO_2_ is the product of cardiac output and the difference between oxygen levels in arterial and in venous blood. The authors conclude that the oxygen carrying capacity of the blood was unaffected, so the reduction in VO_2_ would relate to the ability of the tissues to extract oxygen from the blood. There was a related increase in blood lactate levels; but only a non-significant change in muscle lactate. The authors conclude “These findings question the scientific validity of the current occupational exposure limit for H_2_S”. They do not comment on whether the results are clinically important or related to any likely health effect.

Thus these studies of small numbers of young, healthy volunteer subjects in laboratory conditions were done many years ago by one team of investigators, have not been confirmed by any other research, and show a small number of physiological effects amongst the many outcomes tested. They are inconsistent, with effects seen in one study not seen in others. None of the effects is clearly a health detriment; the clinical significance of the effects seen is unknown. 

#### 3.2.3. Experimental Studies of Humans: Sensory and Cognitive Effects

The ACGIH report cites a study by Fiedler at al. [23], who studied 74 healthy subjects (35 females, 39 males; mean age 25 years) exposed to 0.05, 0.5, and 5 ppm H_2_S. Total symptom severity was not significantly elevated at any dose. Anxiety symptoms were significantly greater at the 5 ppm than at the 0.05 ppm exposure, and related to the odor experienced. No dose–response effect was observed for sensory or cognitive measures. Verbal learning was compromised during each exposure condition. The authors conclude: “Although some symptoms increased with exposure, the magnitude of these changes was relatively minor. Increased anxiety was significantly related to ratings of irritation due to odor.”, and further “Our findings cannot be directly generalized to communities or workers chronically exposed…”, although cautioning that their results in healthy adults could underestimate effects in subjects with other health conditions. 

The ACGIH report also notes an abstract presented in 2000 [24], which assessed symptoms, response to odor, and neurobehavioural performance in 10 subjects exposed to 0, 0.5, and 5.0 ppm. No adverse acute effects or long-term neurobehavioural effects were found. No subsequent publications by this author have been found in the current literature search.

#### 3.2.4. Community Based Studies

The ACGIH report [2] cites several studies as suggesting effects from prolonged low level exposures, but notes that in all these ‘exposure assessments were either lacking or were weak’. They cite a 2002 study from Rotorua, New Zealand [25]; this preliminary study led to more detailed studies published later, as noted below. 

#### 3.2.5. Studies in Experimental Animals

The animal data emphasised in the ACGIH report (page 1) are the ‘minimal changes in the nasal mucosae’ of rats and mice, produced by ‘repeated exposures at 30 ppm, but not 10 ppm’). The basis of the ACGIH limits (page 1) is stated to be upper respiratory tract irritation and central nervous system impairment [2], but the sources cited for the latter relate to exposures at over 100 ppm. 

The studies emphasised most are of ‘subchronic’ exposures. The ACGIH report cites reports (not peer reviewed papers) by Toxigenics, showing no effects below 80 ppm, and therefore a NOAEL (no observed adverse effect level) is the next lower dose tested, 30 ppm. 

Dorman et al. [26] used adult rats (two strains) and mice, and whole-body exposures to 0, 10, 30, or 80 ppm H_2_S for 6 h/day for at least 90 days. There was an increased incidence of olfactory neuronal loss (ONL) after exposures at 30 ppm and above in most groups. Rhinitis was found in mice at 80 ppm H_2_S. In the lung, bronchiolar epithelial hypertrophy and hyperplasia was found at 30 ppm in male and female Sprague Dawley rats, and in male Fischer-344 rats exposed to 80 ppm H_2_S. The authors conclude that “Our results confirm that the rodent nose, and less so the lung, are highly sensitive to H_2_S-induced toxicity, with 10 ppm representing the NOAEL for ONL (olfactory neuronal loss) following subchronic inhalation.” 

However, the authors describe several factors that would make rodents more sensitive to H_2_S than humans [26], noting that “Additional research will be required to determine the significance of the rat and mouse olfactory lesion to an assessment of risk to human health from subchronic exposure to relatively low levels of H_2_S.”

Brenneman et al. [27] exposed 10-week-old male rats via inhalation to 0, 10, 30, or 80 ppm H_2_S 6 h/d and 7 d/wk. for 10 weeks. Following exposure to 30 and 80 ppm H_2_S, there was a significant increase in nasal lesions, limited to the olfactory mucosa. The NOAEL for olfactory lesions in this study is 10 ppm. The authors note three factors that would increase rats’ sensitivity to inhaled exposures more than humans’: rats are obligatory nasal breathers; 50% of their nasal cavity is lined by olfactory mucosa, compared to 5–10% in humans; and rats’ nasal anatomy greatly expands the surface area exposed and slows the air flow. Therefore, they conclude “However, because of differences in the breathing style and nasal anatomy of rats and humans, additional research is required to determine the significance of these results for human health risk assessment.”

They reinforce this caution in another paper [28] comparing lesion locations and regions of high H_2_S flux predicted using a computational fluid dynamics (CFD) model of rat nasal passages. Rats were exposed by inhalation to 0, 10, 30, or 80 ppm H_2_S for 6 h/day for 70 days. At 30 ppm, lesions were only seen in regions lined by olfactory epithelium, and were related to the H_2_S flux. The authors conclude: “These results indicate that airflow-driven patterns of H_2_S uptake within the inherently sensitive olfactory epithelium play an important role in the distribution of H_2_S-induced lesions and should therefore be taken into consideration when extrapolating from nasal lesions in rats to estimates of risk to human health.”

The ‘no effect’ level—NOAEL—reported in these studies is the exposure dose at which no effects were seen (e.g., 10 ppm). However, the no effect level may extend from this up as far as the ‘minimum effect level’–the lowest exposure dose at which changes were observed (usually 30 ppm). The gap between different dose levels tested may be large. Using a computational model shown to be valid for nasal lesions in rats, but assuming the same responses in humans, a NOAEL of 5ppm for humans was predicted [29]. 

### 3.3. Derivation of Exposure Limits in Major Reports

The several expert groups have considered the scientific evidence and recommended various exposure limits (Table 2).

#### 3.3.1. Recommendations of SCOEL

The European SCOEL report [3] provides an assessment of scientific evidence, including the human studies, but has most emphasis on animal experimental studies. Their recommended exposure limits are 10 ppm for short-term 15-min average, and 5 ppm for 8-h time weighted average. 

Their recommendations are mainly based on animal experimental studies of nasal lesions in rats. It is stated (p. 12) that the nasal lesions found in rats are considered the critical effect, and a NOAEL of 10ppm (14 mg/m^3^) is accepted, referring to 4 studies [26,27,28,30]. That is, no effects were seen at this level, the effects being only seen at the next level tested which was three times higher (30 ppm, 42 mg/m^3^).

They say an uncertainty factor relating to species differences is unnecessary. They propose an uncertainty factor of 2 to account for the difference between experimental studies and occupational environment, and “for the limited data set concerning the pathological effects”. In doing so they refer to one human study [18] as showing decreased oxygen uptake and increased blood lactate. 

Using this factor of 2, they set the longer term exposure limit at 14/2, that is, 7 mg/m^3^, 5 ppm. They set the short-term limit at twice this, 14 mg/m^3^, 10 ppm.

These factors are questionable. The species differences between rats and humans would tend to make rats more sensitive, that is, effects on rats may be seen at lower exposure dosages. The detailed comments by the investigators in these studies, reviewed earlier in this report, are not discussed. The phrase “the limited data set concerning the pathological effects” is not explained or further discussed. They cite one human study [18] and refer to a significant decrease in oxygen uptake with an increase in blood lactate after short-term exposure. The study cited in fact showed a significant increase in oxygen uptake; a further study by the same investigators showed no effect [19], while it was a further study which showed a decrease in oxygen uptake [21]. The inconsistency of the studies and the small numbers of subjects involved are not mentioned.

#### 3.3.2. DECOS 

The DECOS recommended limits are substantially different from those of SCOEL [5].

They use nasal lesions in rats found after exposure as the critical effect, referring to 3 studies, and using a NOAEL limit of 10 ppm. They regard an extrapolation factor to compensate for species difference unnecessary (again, without discussion of the issues raised by the authors of the animal studies). To account for differences in exposure pattern between the experimental and the occupational setting, and for the limited data set concerning the pathological effects, they propose a factor of 2. Thus far, their arguments are identical to those of SCEOL. 

They then state that “differences among people should be taken into account”, and propose a factor of three for this. They note two groups that could be more sensitive, each based on one study: subjects with asthma could be susceptible at levels as low as 2 ppm [13], pregnant women might have an increased risk of spontaneous abortion [17]. That gives a factor of 2 × 3 = 6, compared with 2 from SCOEL. The DECOS recommended 8-h limit is therefore 14/6, which is 2.3 mg/m^3^, 1.6 ppm. They make no recommendation about a short-term limit.

The difference between the limits of SCOEL and DECOS is therefore not in the scientific information reviewed, nor in the interpretation of it, and both groups use an uncertainty factor of 2 on the basis of differences in setting and limitations of pathology. The differences arise because the DECOS group adds an additional reduction factor of 3 to allow for ‘differences among people’.

#### 3.3.3. ACGIH

The derivation of the ACGIH recommended limits is less clear. The scientific studies reviewed by ACGIH are basically the same as those reviewed by SCOEL and DECOS; only the Fiedler study [23] is not considered in the European reports. Despite this, ACGIH comes up with recommendations which are considerably lower in exposure values than the other reports. There are no clear reasons for this discrepancy discussed in the ACGIH documentation.

For the human studies, it stated that metabolic changes were seen at 5 ppm (citing [18]), although the changes were regarded as not clinically significant by the investigators, and similar changes were not seen at 5 ppm in a later study from the same investigators [18]. The ACGIH report states (p. 1) that “changes were not considered clinically significant but were seen at exposures less than 5 ppm”. This statement is unjustified. Only one of these studies did include 0.5 and 2.0 ppm exposures [18], and only one *p* < 0.05 significant change was reported in 57 comparisons made at these dosages. 

For human studies, ACGIH concludes that 2 ppm is the NOAEL level and 5 ppm the LOAEL (lowest adverse effect). They justify this by referring to [18], saying “There were trends in the data suggesting the beginning of a dose response relationship. At 2 ppm VO_2_ and blood lactate increased, while carbon dioxide output VCO2 decreased.” However, none of these changes was statistically significant, and to make this interpretation from a study of 16 subjects seems to go beyond the data. Further, while [18] reported an increase in VO_2_ at 5 ppm, these investigators later reported no change at 5 ppm [19], and in their most recent study reported the opposite effect, a decrease in VO_2_, at 10ppm [21]. ACGIH also cite the study by Fiedler [23] which showed an increase in symptoms of anxiety at 5 ppm, but this was significantly related to irritation due to odor. 

For animal studies, they cite two of the studies in rats on nasal effects at 30 ppm, and also a study of metabolic changes in animals at 20 ppm [31] which is not given prominence in the other reports. Unlike the other groups, they do not specify uncertainty or safety factors, or state what NOAEL level they are assuming. Later in the document in discussing animal experiments the same NOAEL level of 10 ppm is accepted. 

Following this they simply state their recommendation that a 1 ppm longer term limit “should be sufficient to protect against all the unwanted effects of hydrogen sulphide”. They note that peak exposures of 5ppm “would not be expected to produce more serious effects on the respiratory, central nervous, or cardiovascular systems” and so recommend 5 ppm as a short-term limit. 

### 3.4. Human Studies Published since 2006 and Not Considered in These Reports

#### 3.4.1. Occupational Studies of Chronic Exposures

There have been several occupational studies published since the major reviews. However, none of these has clearly shown specific health effects related to a reliably documented dose of H_2_S without likely contamination with other exposures. The interpretation of these studies in terms of exposure limit guidelines is therefore limited.

A study of wastewater treatment plant workers in Iowa compared them to water treatment plant workers regarded as unexposed [32]. Of 140 wastewater workers, 93 (66%) completed questionnaires, and 67 (48%) participated in the H_2_S exposure monitoring studies. Exposure levels of H_2_S in all monitored tasks were under 1 ppm. Endotoxin levels were also monitored. Increased respiratory, ocular and skin irritation, neurological, and gastrointestinal symptoms were recorded by wastewater workers. Tasks involving sludge handling and plant inspection were associated with memory/concentration difficulties, throat irritation, and stomach pain. 

In Norway, a large study of farmers with measurement of many exposures at 127 randomly selected farms showed increases in chronic bronchitis and chronic obstructive pulmonary disease associated with levels of H_2_S, but also of ammonia and of inorganic and organic dusts. The effects of different exposures could not be separated [33]. 

In Iran, blood parameters for 110 workers in natural gas processing plants, with exposures of 0–90 ppb H_2_S for 1–30 years, were compared to those of 110 unexposed males [34]. The mean methemoglobin was higher, and sulfhemoglobin lower, in the exposed group, but neither showed any trend with years of exposure. The authors misinterpret their own data: they state in the discussion and abstract that sulfhemoglobin was increased, while their results, in text and tables, show it was reduced. The authors interpret their study as showing potential harmful health effects, but it has little credibility.

In Egypt, urinary thiosulphate was used as a marker of H_2_S exposure, and clinical neurological examinations and cognitive tests were applied to 33 male exposed sewage workers and to 30 comparison subjects [35]. Neurological symptoms were increased and performance on most neuropsychological tests was poorer in the exposed workers, indicating cognitive impairment.

In a study of 34 male oil field workers in Iraq seen at an occupational clinic, whose symptoms were clinically assessed as related to H_2_S exposure, the most common symptoms were bleeding from the nose, pharynx, gum or mouth; no asthmatic attacks were reported. The levels of exposure were unknown and this seems likely to relate to high, accidental exposures [36]. 

In 56 fishermen assessed in China, 46 had eye burns; they came from 21 vessels, with H_2_S concentrations in the boats’ cabins of 99 +/− 38 mg/m^3^ [37]. H_2_S concentrations in the boats before the caught fish were discharged were 219 +/− 31 mg/m^3^, dispersing by 90% after 1 h in boats with ventilation and spraying but by only 18% in those using only natural ventilation. The threshold concentration that could cause the eye burns was estimated as 38 mg/m^3^, with an exposure time of 120 min (paper in Chinese, English abstract only reviewed).

In Poland, groups of copper miners working in areas with no H_2_S emissions recorded in the previous 10 years (*n* = 237), and in areas with H_2_S emissions recorded (*n* = 88) were compared [38]. There were no findings typical of H_2_S toxicity, and no detectable sulfhemoglobin. The exposed group tended to have higher serum levels of enzymes involved in the synthesis of H_2_S, cystathionine beta-synthase and cystathionine gamma-lyase, and higher activity of angiotensin converting enzyme. The objective was to study levels of the enzyme for H_2_S toxicity, cytochrome C oxidase (COX), and decreased levels were found: the authors suggest COX may be an indicator for H_2_S exposure (paper in Polish, English abstract reviewed).

#### 3.4.2. Community Studies of Effects of H_2_S Exposures

The most detailed community studies, involving individual health assessments and exposure estimates based on extensive monitoring, are from Rotorua, New Zealand, which has the world’s largest population exposed to ambient H_2_S from geothermal sources. The levels of exposure are much lower than occupational exposure limits; but these exposures are 24 h, every day, and can be for life. Long-term monitoring results show ambient levels of up to 110 ppb, equivalent in total dose terms to a working week level of 460 ppb [6]. In these studies, adult residents aged 18 to 65 were enrolled during 2008 to 2010. Residences, workplaces and schools attended over the last 30 years were geocoded. Exposures were estimated from data collected by summer and winter H_2_S monitoring networks across Rotorua. Four metrics for H_2_S exposure, representing current and long-term (last 30 years) exposure, and time-weighted average and peak exposures, were calculated. The maximum exposure group (top quartile) had an exposure level by time weighted average over 30 years, of 19 to 58 ppb; and maximum exposure levels of 32 to 60 ppb [39].

These studies have shown no detrimental health effects. For lung function, chronic obstructive pulmonary disease and asthma, 1204 subjects had spirometry studies [39]. The study found no evidence of reductions in lung function or increased risk of COPD or asthma, from recent or long-term H_2_S exposure. Indeed, there were results suggesting improved lung function associated with recent ambient H_2_S exposures. An earlier study of 1637 subjects also showed no associations with self-reported asthma or asthma symptoms [40]. 

For eye disease, 1637 adults ages 18 to 65, underwent a comprehensive ophthalmic examination to assess nuclear opacity, nuclear colour, and cortical and posterior subcapsular opacity [41]. No associations were found between estimated H_2_S exposures and any of the four ophthalmic outcome measures, including cataract. The authors conclude that an association reported previously based only on hospital discharges [25] was due to limitations of its ecological study design. 

A further study gave no evidence that H_2_S exposure increases risks of peripheral neuropathy [42]. In a detailed study of cognitive function of 1637 adults, neuropsychological tests were used, measuring visual and verbal episodic memory, attention, fine motor skills, psychomotor speed and mood [43]. There was no impairment of cognitive function; indeed, higher exposures were associated with faster response times and better performance on some tests. This may show a beneficial effect to prevent degradation of neurons related to Parkinson’s disease [44].

Only earlier, weaker, ecological studies in Rotorua were included in the ACGIH report, one suggesting associations of H_2_S exposure with nervous system, respiratory and cardiovascular diseases and cataract [25], and another study using hospital discharge data suggested associations with chronic respiratory diseases [45]. 

Several studies from other areas are also ecological studies. In Reykjavik, Iceland, from 2003–2009, daily deaths from all natural causes and from cardiovascular disease were compared to daily ambient concentrations of H_2_S, with various lag periods [46]. Positive associations were found in the summer, in males, and in the elderly; adjustments were made for traffic-related pollutants and meteorological variables. The authors note that “… the results should be interpreted with caution. Further studies are warranted to confirm or refute whether H_2_S exposure induces premature deaths.”

In a related study in Reykjavik, from 2007 to 2014, daily area-level H_2_S exposures from a geothermal power plant were modelled, and compared to the numbers of emergency hospital visits [47]. Increases in visits for heart disease were seen with levels of over 7.0 micrograms/m^3^, more pronounced in males and in those over age 73; no associations were found with other diseases. Further studies showed associations between daily ambient levels of H_2_S (and also particle pollution, PM_10_) and dispensing of anti-asthma drugs [48]. A similar study showed associations between dispensing of anti-angina medications with levels of nitrogen dioxide and of ozone, but not of H_2_S or particulates [49]. 

Several studies compare people living in areas assumed to have higher exposure to H_2_S, and other pollutants, with residents in other areas. In Iceland, a comparison of cancer incidence in residents of a high-temperature geothermal area with residents of a cold, non-geothermal area showed excesses of all cancers combined and of several specific types of cancer; the results were adjusted for age, gender and social status, but other confounders could not be taken into account [50]. In contrast, in Alberta, Canada, no excess of all cancers, apart from cervical cancer, was found in First Nations people living near sulphur-rich natural gas areas [51].

Several studies related to the Kilauea volcano in Hawaii showed increases in respiratory and various other acute illnesses, but likely related to sulphur dioxide and sulphur particles [52,53,54,55]. In the Azores, Portugal, health centre records of chronic bronchitis showed higher rates in a volcanically active area than in a comparison area, with exposures to H_2_S and sulphur dioxide noted [56]. Likely H_2_S exposure has been associated with asthma symptoms in the United Arab Emirates [57] and in Vladivostok, Russia [58], but in situations where many pollutants were involved.

In Beijing, China, in a small study of patients with chronic obstructive pulmonary disease (23 subjects), increased daily air pollution was associated with increased biomarkers and with symptoms [59].

Several small studies have shown health effects in residents near sources likely to produce H_2_S, but these effects may be related to odour or to other emissions. In New Mexico, USA, 49 subjects likely to have been exposed to community levels of H_2_S from a sewage plant, oil refineries, natural gas or oil wells, and a cheese-manufacturing plant and its waste lagoons, were assessed, and compared to 42 unexposed subjects from Arizona [60]. Very many tests were done, and the authors concluded that neurobehavioural functions were impaired in the exposed subjects. In response to community concerns in Dakota City, Nebraska, USA [61], neurobehavioural tests were administered to 171 residents in an area with reported high H_2_S exposures (over 0.09 ppm) and 64 residents in a comparison area with less than 0.05 ppm; no significant differences were seen. 

Health and mood symptoms were related to odor in 23 residents near landfills in North Carolina who kept diaries, and odour increased with higher H_2_S levels recorded, with a maximum of 2.3 ppb [62]. Acute eye irritation increased with H_2_S levels in 101 adults living near swine feeding operations in North Carolina, and other symptoms were associated with particulate levels and odour [63]. Similar effects in 25 subjects near animal (hog) feeding operations in Ohio [64] were attributed to H_2_S, but without measurements. 

In a geothermal area in Tuscany Italy [65], residents from 1980 to 2016 in six municipalities (*n* = 33,804) were categorised into three groups by estimated average exposure to H_2_S (<5, 5–20, and 20+ µg/m^3^), and linked to information on deaths up to 2014 and hospital discharges up to 2016. Comparisons were done between exposure categories, by sex and in total, and with a dose trend analysis, with 19 major categories of death and of hospitalisation; so 266 comparisons are presented. Due to the large volume of data, all the differences, even if statistically significant, are small in absolute terms. For mortality, there were small but significant excesses in deaths from diseases of the respiratory system (12% increase per 7 mcg/m^3^) and, within that category, for pneumonia; there were significant decreases in total non-accidental mortality, total cancers, ischaemic heart disease, and myocardial infarction (25% decrease). For hospitalisations, there were significant increases in pneumonia and in chronic obstructive pulmonary disease (COPD), but a significant decrease in other respiratory diseases, giving no significant difference in total respiratory hospitalisations. There were significant increases in hospitalisations for heart failure, diseases of the circulatory and nervous systems and diseases of the ovary; and a significant decrease for all cancers. The authors emphasise respiratory disease in their summary, claiming consistency between mortality and morbidity data, but the excess in mortality for total respiratory diseases was small and marginally significant (RR 1.12, 95% confidence limits 1.00–1.25); and there was no increase in hospitalisations (RR 1.02, CL 0.98–1.06).

Bustaffa et al. [66] have reviewed health studies done in communities living near geothermal plants producing heat and electricity, including studies in New Zealand, Iceland and Italy. Most of these studies are ecological, or are cohort studies in which exposure is estimated the basis of residence. The studies are reviewed elsewhere in the current paper. They point to many inconsistencies in the results, with some evidence of exposure to low concentrations of H_2_S being associated with increased respiratory disease, but exposure to high levels associated with a decrease in cancer mortality. They regard the studies as limited because of their ecological design and inadequate exposure assessment.

A large review of health effects of hazardous waste [67] considered 95 health outcomes, and reported ‘sufficient evidence’ of associations of exposure to oil industry waste that releases H_2_S with a wide range of health conditions. While such wastes may have H_2_S emissions, these are mentioned specifically only with reference to acute effects related to illegal dumping in Abidjan [68] with high exposures, and there organo-chloride compounds, mercaptans and other agents were also present. For the many other associations, exposures including benzene, chlorinated phenyls, dioxins, heavy metals, and endocrine disrupting chemicals, are considered. In an economic assessment, H_2_S from oil and gas fields has been assessed as causing considerable health costs in Kazakhstan [69]. 

#### 3.4.3. Studies on Mechanisms

In addition to its established effects, H_2_S has recently received much interest as a physiological signaling molecule involved in a number of physiological processes, and shows some cytoprotective effects [70,71,72]. It is suggested the brain effects may be beneficial at low exposures and detrimental at high exposures [73]. Similarly, effects on inflammation may be beneficial or detrimental [74,75]. 

## 4. Discussion

The recommendations from ACGIH are for limitations of occupational exposure well below the minimum level at which any effect has been documented. Such limits should indeed protect against health effects; but whether they provide any further protection than do current limits is questionable.

Setting appropriate exposure limits for occupational exposure, or for public exposure, also requires consideration of other issues. These include the feasibility of restricting human activities; the costs of doing so; the logistics and costs of measuring exposure levels accurately and routinely; and any detrimental effects of adhering to such limits. For instance, if the limits meant that workers had to use breathing equipment for some tasks that could involve other risks.

On the basis of the available scientific evidence, there is no firm evidence that exposures up to current occupational exposure levels cause any detrimental health effects. Eye irritation, after exposures to both H_2_S and CS_2_ has been seen; but no eye or respiratory effects have been seen in studies of H_2_S exposure alone below existing limits. 

Although exposed at lower levels, long-term, indeed lifetime, exposure to unusually high ambient levels of H_2_S is accepted in communities such as Rotorua, New Zealand. Extensive high-quality studies of individuals in this community have assessed respiratory diseases including asthma, eye disease, peripheral neuropathy, and intellectual function, amongst other outcomes, and have shown no adverse effects. Earlier ecological studies in Rotorua had suggested several positive associations, not confirmed by these more detailed studies. Thus, the ecological studies in Iceland and Italy showing associations with various measures of disease require verification from more powerful studies. There are several small studies showing associations which also need validation, and most of these cannot separate effects of H_2_S from those of other pollutants.

Three expert group reports have been reviewed, from SCOEL, DECOS, and ACGIH. All of them base their recommendations primarily on respiratory and related effects in human volunteers under laboratory testing, and on studies of H_2_S exposure on the nasal passages in rats and mice. The documentation for the German MAK (maximum workplace exposure) levels [76] is generally consistent with these reviews. It is based on the same literature, and provides a more detailed review of the human volunteer studies than the other reviews. A NOAEL from animal studies of 10 mg/m^3^ is accepted. This group notes that considerable odour annoyance is to be expected at exposures at 5 mg/m^3^, although they state that it is not possible to assess if this interferes with occupational activities or produces adverse effects. On this basis they recommend a MAK value of 5 mg/m^3^. 

The human studies were studies of small groups of young healthy volunteers by one investigator group, and conducted over 20 years ago. They have not been further validated or independently tested. The studies assess a large number of physiological parameters, all of questionable clinical importance. The effect most often emphasised, a decrease in oxygen uptake, was seen in one study, but not seen in one previous study, and the opposite effect was initially reported. The reviews by all three expert groups pay insufficient attention to the inconsistency of these effects, their small sizes, the lack of reproducible findings, and their questionable clinical relevance.

For the studies of rats and mice, the extrapolation of these results to humans exposed at the same levels of exposure is questionable. The effects seen are reported only at exposures of 30 ppm and above. The investigators emphasise that the breathing style and nasal anatomy of rats is very different to humans and would make these animals much more sensitive to inhaled agents. The three review groups take no consideration of species differences. The threshold for any effect in these studies would be between 10 and 30 ppm. In formulating recommendations, the SCOEL group use a reduction factor of 2 for long-term exposure; the DECOS group uses a reduction factor of 6; and the ACGIH group does not state how its recommendations are derived, but effectively use a reduction factor of 10. These factors are not clearly justified. The derivation and use of uncertainty factors in deriving occupational exposure limits varies considerably; in a review of 128 recommendations from SCOEL, uncertainty factors were explicit in 50% of reports, and decreased over time from 64% in 1991–2003 to 37% in 2004–2017 [77]. When explicit, the factors were 1.8 times larger than the comparative implicit safety margins in other reports.

This review shows that the substantial differences in occupational exposure limits given by three important scientific groups are not due to differences in the scientific evidence the groups have reviewed. With minor exceptions, the three groups are relying on the same scientific studies, and to a large extent they agree on the interpretation of the individual studies. The differences in the recommended exposure limits arise from differences in the application of uncertainty factors. In a review of the scientific basis of using uncertainty factors, Dankovic et al. [78] note that such factors have been used since at least 1954, and point out that their application varies greatly between organisations. In studies they reviewed, they concluded that none presented adequate scientific support for the uncertainty factors used. They note that “the use of uncertainty factors is a scientific judgement based on a policy decision to determine a safe concentration or dose”. They describe the scientific rationale for the use of several specific uncertainty factors to deal with issues such as extrapolating animal to human data, considering sensitive sub-populations, considering short-term and long-term exposures, and the interpretation of no effect and lowest observed effect levels. However, they conclude that the process is improving and now “consideration of uncertainties in occupational exposure limit derivation is a systematic process whereby the factors applied are not arbitrary, although they are mathematically imprecise”. They call for recommendations for exposure limits to be fully explicit in terms of the uncertainty factors used and how they are combined. This current review of the situation with one chemical reinforces this view, showing that differences in the application of uncertainty factors of the main source of variation in the exposure limits recommended by different groups, and that the documentation of how these factors have been derived and used may be deficient.

The review groups appear to be taking a precautionary approach by accepting inconsistent evidence, down-playing limitations of studies even when these are stressed by the original investigators, and including various reduction factors. Thus, they recommend as occupational exposure limits, levels at which no health effects, even innocent physiological changes, have ever been demonstrated; these levels are considerably below the minimum levels at which any detrimental health effect has been shown. Therefore, the scientific justification for these recent recommendations of lower occupational exposure levels is questionable. There is no clear evidence that reducing limits to follow these recommendations will protect the health of workers any further than will the current exposure limits used in most countries.

These expert reports were done nearly 10 years ago, and there is considerable relevant scientific literature published since then. Further, relevant information may be in gray literature such as government and company reports. A full review, involving a multidisciplinary expert group and considerable resources, would be appropriate.

## 5. Conclusions

In recent years, three expert groups have used essentially the same scientific evidence to recommend different occupational limits for exposure to hydrogen sulphide gas. All these differing levels are lower than currently accepted occupational exposure limits in most jurisdictions; so, industrial workers have been exposed for many years to levels of exposure likely to be higher than these new recommended low limits, while being within the existing limits. The new lower limits are justifiable only if these exposures produce a health hazard.

In the scientific evidence, the human volunteer studies were done in one laboratory over 20 years ago, and give inconsistent results. The animal studies are of animals whose breathing style and nasal anatomy would make them much more sensitive than humans to inhaled agents. No clear evidence of detrimental health effects in workers has been found. Studies of lifetime exposure in communities with natural sources have shown no detrimental effects.

A limited review of studies published since 2006 has shown several occupational or community studies, but these show no confirmed hazards. A further review of the relevant scientific evidence is needed to support any major changes in occupational exposure limits. The current information is of course limited, particularly in regard to the effects of human exposures within the limits of the highest current occupational exposure limits and the suggested lower limits. Further research may clarify the situation.

This review suggests that expert groups have paid insufficient attention to the inconsistency of these effects, their small sizes, and their questionable relevance to human health. Each expert group applies different reduction factors without clear justification. The conclusion is that the scientific justification for lowering occupational exposure limits is very limited. There is no clear evidence, based on currently available scientific evidence, that lower limits will protect the health of workers further than will the current exposure limits used in most countries. 

## Figures and Tables

**Table 1 ijerph-18-02866-t001:** Occupational exposure levels for hydrogen sulphide currently in 31 jurisdictions.

Jurisdiction	Limit Value-Eight Hours		Limit Value-Short Term		Notes
	ppm	mg/m^3^	ppm	mg/m^3^	
Australia	10	14	15	21	
Austria	5	7	5	7	
Belgium	5	7	10	14	
Canada-Ontario	10		15		A
Canada-Québec	10	14	15	21	
Denmark	10	15	20	30	
European Union	5	7	10	14	F, A
Finland	5	7	10	14	A
France	5	7	10	14	B
Germany (AGS)	5	7.1	10	14.2	A
Germany (DFG)	5	7.1	10	14.2	A
Hungary		14		14	
Ireland	5	7	10	14	C
Italy	5	7	10	14	
Japan	10				
Japan-JSOH	5	7			
Latvia		10			
People’s Republic of China				10	D
Poland		7		14	
Romania	5	7	10	14	A
Singapore	10	14	15	21	
South Korea	10	14	15	21	
Spain	1		5		
Sweden	5	7	10	14	A
Switzerland	5	7.1	10	14.2	
The Netherlands		2.3			
Turkey	5	7	10	14	A
USA-NIOSH			10	15	E
USA-OSHA			20		
United Kingdom	5	7	10	14	

Notes: A = 15 min average value; B = Restrictive statutory limit values; C = 15 min reference period; D = ceiling limit value; E = 10 min ceiling limit value; F = (EU) Indicative Occupational Exposure Limit Values and Limit Values for Occupational Exposure Binding Occupational Exposure Limit Value–BOELV. GESTIS International Limit Values. From [10].

**Table 2 ijerph-18-02866-t002:** Recommendations for occupational exposure limits for hydrogen sulphide.

Advisory Body	SCOEL	DECOS	ACGIH
**Date**	**2007**	**2006, 2010**	**2010**
Conclusions from literature			
animals, no effect level	10 ppm	10 ppm	10 ppm
humans, start of effects			5 ppm
Uncertainty factors			
species differences		3	unspecified
exposure duration and limited pathological data	2	2	
total	2	6	unspecified
Recommendations			
8-h exposure limit	5 ppm	1.6 ppm	1 ppm
short term exposure limit (STEL)	10 ppm		5 ppm

Sources: [2,3,5,11]; SCOEL: Scientific Committee on Occupational Exposure Limits; DECOS: Dutch Expert Committee on Occupational Standards; ACGIH: American Conference of Government Industrial Hygienists.

## Data Availability

Not applicable.

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
