# Peer review of "The Scientific Basis for Occupational Exposure Limits for Hydrogen Sulphide—A Critical Commentary"

_ijerph, 2021, doi:10.3390/ijerph18062866_

Round 1

Reviewer 1 Report

This paper attempts to review the current state of knowledge of the levels and concomitant effects of hydrogen sulfide (H2S) exposure and to determine if there is currently sufficient evidence to lower exposure limits based on that information. The author concludes there is not sufficient evidence to warrant reduction. The author does cover, to varying degrees, the pertinent literature on this topic.

I have a problem with the tone of the paper, on occasion, and the handling of the synopses of the studies. The author appears to be a little overzealous in downplaying findings that might disagree with what appears to be a pre-determined conclusion that the evidence for lowering a standard will be found to be insufficient. Obviously, the author should have started writing the paper with his own conclusion firmly in mind. Just as obviously, the author is going to conclude that the evidence is insufficient, as stated in the abstract.

However, there needs to be more attention paid to possible positive findings, whether they present a consistent, coherent picture of a syndrome of any sort and then whether it is the symptoms or the experimental techniques that may be lacking in supporting the lowering of the standard. The two seem to often be muddled in the author’s exegesis and too summarily dismissed for varied and inconsistent reasons.

As I was reading through the paper I thought I detected the possibility, whether the author intended it or not, that there might be an inflammatory mechanism at work in many of the reports as the author related them. Most of the papers being cited do not seem to mention that possibility. It is no wonder, then, that the author would not bring it up, except that it receives a brief mention in section 3.4.3.

The references 71 and 72 are noted in relation to that hypothesis in a single sentence and the paper goes on with no further note. It made me wonder whether the author had so convinced himself of his conclusion by that point that a potential and “newer” hypothesis explaining the mechanism and potentially arguing for a diminution of the exposure limits due to that mechanism might not have been in some of the other papers that were dismissed earlier. Certainly, there were reference to cardiovascular effects, specifically reference 25, 48 and 49. Cardiovascular effects are certainly seen in a myriad of studies associated with inflammation.

While I do not disagree that the older literature is not strongly supportive of lowering standards, I would disagree with the strong negative tone of the actual conclusion, which states, “On the basis of the available scientific evidence, there is no firm evidence that exposures up to current occupational exposure levels cause any detrimental health effects.” It certainly appears from just brief references that potential evidence may exist but has not yet been fully investigated.

Editorial Note: The American Conference of Governmental Industrial Hygienists is the ACGIH, please correct any usage of "ACHIG" to be in line with that.

Reviewer 2 Report

The author deals an interesting topic, maybe more occupation than public health issue but it could be considered of general interest in a large view.

The information reported in it is quite familiar for everyone working in this field, the literature is quite covered and well reported.

There is low interest for the publica health whereas there is a large interest for the occupational issue. 

In any case, I suggest to add a section regarding the personal protective equipments (PPEs), so the issue is totally covered. Please, also revise all the underscripts which are hard mistakes (e.g., H2S, VO2, etc.).

Hydrogen sulphide must be written with 2 underscript throughtout the manuscript as well as carbon disulfide: please change in the manuscript.

Reviewer 3 Report

The paper reads well and gives a critical commentary of the scientific basis for occupational exposure limits for hydrogen sulphide. Although the author tries hard to be balanced his assessment of the evidence is evident by the tone of language used. It is clear that scientific evidence for the hazard associated with H2S is weak and this is at least acknowledged by SCOEL. For example, they repeatedly wrote, “there is limited information” or “no data are available”  in relation to various areas of the dossier. The safety factors applied by the agencies are not arbitrary (see https://www.ncbi.nlm.nih.gov/pmc/articles/PMC4643360/) but equally there is a considerable amount of judgement involved in their application. One important issue is that most of the human experimental evidence has been collected on healthy (young?) volunteers and this may not be entirely representative of the working population. DECOS discusses the possible added risk for people with asthma, and this may be an important aspect of their recommendation for a lower limit than SCOEL.

The methods section is inadequate and contains material that should more correctly be in the results section, i.e. number of citations screened etc. There are no details of what bibliographic databases were searched, the search terms or the criteria used for identifying the papers for review.

I would like to see more information about the stimulus for the research, which seems to be the introduction in New Zealand of an interim limit of 5ppm (8-hr) and STEL of 10ppm, with proposed reduction to change to a WES-TWA of 1ppm and WES-STEL 5ppm in the year 2022. What was the arguments from the government authorities to sustain their judgement? Why is the change introduced in two stages – are there sectors that can’t comply immediately? The introduction could also have some more detail of the SCOEL and DECOS recommendations, which have subsequently been endorsed by the relevant regulatory agencies.

Although there was no declared financial contribution to the writing of the paper it is perhaps disingenuous to write there was no external funding for the work because, as I understand, a company paid for the review of the scientific literature.

Minor comments

The aurthor does not mention the German MAK value documentation (2013), which seems relevant - https://onlinelibrary.wiley.com/doi/epdf/10.1002/3527600418.mb778306e4313

There are numerous typographical errors that must be corrected, e.g. H2S should have the number subscripted

Page4: “ccould”

Page 4: “exercise.Because”

Page 5: “..”

Page 7 mg/m3 – number should be superscripted

Page 8: ACHIG should be ACGIH

Page 8 VO2 – subscript number

Page 9: to110 ppb

Page 10: “…function; . indeed…”   - ACHIG again

Page 11: “3.1 Review…” – missing section?

Page 11: CS2 – subscript number

Round 2

Reviewer 1 Report

Minor editorial corrections are:

p11.  Dakota city should be Dakota City

p12. 10 ml/m3 and 5 ml/m3 (used twice) should be 10 mg/m3 and 5 mg/m3

p14. "...no clear evidence, on currently available.." might be better phrased as "....no clear evidence, based on currently available..."

Reviewer 2 Report

The authors made a good effort for improving the manuscript, he introduced an important section regarding the occurrence in other situations, e.g. Tuscany, he corrected some mistakes, now it could be considered for publication.